# Development and Validation of a Scale Measuring Intention toward Participating in Pro Bono of Pre-Service Physical Activity Instructors for the Activation of Physical Activity for the Disabled: Based on the Theory of Planned Behavior

**DOI:** 10.3390/healthcare10102094

**Published:** 2022-10-20

**Authors:** Kyungjin Kim, Yonghwa Lee

**Affiliations:** Department of Adapted Physical Activity, Korea Nazarene University, Cheonan-si 31172, Korea

**Keywords:** pro bono, intention, pre-service physical activity instructor, people with disabilities, theory of planned behavior

## Abstract

The purpose of this study was to develop and validate a scale for predicting the intention of pre-service physical activity instructors for persons with disabilities to participate in pro bono work, based on the Theory of Planned Behavior. This study analyzed 322 university students majoring in adapted physical activity in South Korea. To determine the purpose of the study, the EFA using SPSS 21.0 and CFA using AMOS 21.0 were used to confirm the validity of the measurement tool and the relationship between latent and observed variables. Further, the Cronbach’s alpha was used to identify the internal reliability. As a result, first, the questionnaire used in this study was validated based on the theory. Second, the behavioral belief was influenced by teaching experience about physical activity for the disabled and knowledge about physical activity for the disabled. Third, the normative belief was influenced by the parents of people with disabilities, people with disabilities, family members, friends, and students in my department. Fourth, the control belief was influenced by the state of mind of physical activity instructors for people with disabilities, the ability to create an IEP, and the ability to do physical activity.

## 1. Introduction

The population of persons with disabilities registered in 2021 was about 2,645,000, which has been maintained at about 5% of the total population of South Korea for 10 years, since 2010 [1]. An increase in the disabled population is the same as a decrease in the non-disabled population across the Organization for Economic Cooperation and Development (OECD) [2]. The number of disabled elderly people is increasing due to the increase in the elderly population, and the amount of people with birth defects is also increasing [3]. About 60% of disability occurs in old age [1]. As for the status of children with disabilities, the number of children with disabilities in the total child population in 2008 has been continuously increasing from 0.73% to 0.89% in 2018 [1]. Although the proportion of the population with disabilities is maintained or increased, the reality is that the level of health, which is significant for the daily life of the disabled, is not as good as that of the non-disabled. For example, 50.2% of the disabled reported the subjective health status of the disabled as bad [4]. In the case of chronic diseases, the prevalence of hypertension in the non-disabled adult group was 33.5% and the prevalence of diabetes mellitus was 13.0%. However, in the case of the disabled, hypertension was 46.9% and diabetes was 21.9%, which was higher than that of the non-disabled [5].

The importance of exercise is being emphasized to improve the health status of individuals with disabilities. Exercise maintains and improves physical strength and body functions through physical training, so it can be effective for health management as well as disease prevention [6]. When asked the purpose of physical activity for people with disabilities, 50.0% answered health promotion and management, and individuals with disabilities recognized that physical activity is important for the health of the disabled. However, in the case of the disabled, not only participation in sports but also the will to participate is shown to be low [7]. In the case of non-disabled people, it can be seen that 52.4% of them participate in physical activity more than twice a week, but for people with disabilities, that figure is 23.8%. Among those with no experience in physical activity, the amount of individuals with a view of “I have no intention of doing sports at all” or “I have no intention of doing it at all” is as high as 50.0% [7]. The reasons for not participating in physical activity for the disabled include cost, lack of sports facilities for the disabled, and the need for exercise equipment and auxiliary personnel (help for transportation, etc.). Although these problems are being improved, persons with disabilities point out the lack of life sports instructors for the disabled in regard to complementing the diversity of programs [6,7].

Currently, the Korean government is planning to approximately double the number of physical activity instructors for the disabled from 577 in 2018 to 1000 in 2022 [7]. This might be evidence that the current government is seriously aware of the lack of physical activity instructors for the disabled. However, it has been determined that more physical activity instructors for the disabled are needed to improve the quality of sports for the disabled and the continuously increasing proportion of disabled people in South Korea [8]. Therefore, pro bono work has been recently proposed as a method to solve such a problem. Pro bono, a shortened version of pro bono publico, is a Latin phrase that means “for the public good”. Experts do not seek compensation based on their expertise, but provide material and mental help for the benefit of society, either directly or through charitable organizations [9]. The implementation of pro bono work can solve the shortage of physical activity instructors for the disabled and can help provide various programs.

The concept of pro bono work started with American lawyers providing legal services for the socially disadvantaged, and experts providing their knowledge and services free of charge [10]. Pro bono is an area of volunteer work, and provides services with respect to individual abilities, knowledge, experience and skills. Along with talent donation, it has recently established itself as a new trend in social contribution roles [10]. However, talent donation is short-term skilled volunteering that targets individuals as a form of volunteer activity or donation that provides one’s professional abilities and skills to society, whereas pro bono, which also provides professional skills and is defined as a volunteer activity, offers services not only to individuals but also to organizations such as non-profit organizations and social enterprises [11]. In particular, pro bono work is not a short-term activity such as volunteering or talent donation, but a long-term social contribution activity. Pro bono is currently expanding not only to legal services but also to business, arts, marketing, education, and medical care across the world [12].

Most of the Korean and international studies related to pro bono work are the establishment of the concept of pro bono and the study of pro bono activation methods, and further research is currently being conducted [10,11,12,13,14]. Studies of pro bono related to law, education, design, and beauty are being conducted, but the reality is that there are no prior studies related to physical activity for the disabled [10,15,16]. In particular, studies on physical activity for the disabled through volunteer activities have been conducted, but there are no studies related to pro bono work, so research is needed to promote physical activity for the disabled through pro bono work [8,17,18,19,20,21].

In addition, theories based on various factors are being used in relation to physical activity for the disabled, including Theory of Planned Behavior, Theory of Reasoned Action, Health Belief Model, and Social Cognitive Theory. The Theory of Planned Behavior is one of the theories most widely used in relation to physical activity or sports for the disabled [22,23,24,25]. The Theory of Planned Behavior was developed based on the Theory of Reasoned Action of Fishbein and Ajzen [26], and it states that an individual’s specific behavior is determined according to the intention to perform. The Theory of Planned Behavior is a theory that supplements the problem that the individual’s control over a specific behavioral intention is excluded from the Theory of Reasoned Action, and it expresses the factors of an individual’s attitude, subjective norm, and control toward a specific behavioral intention [26,27]. Thus, the Theory of Planned Behavior can predict specific behavior according to behavioral intentions [27]. Hence, if the intention of physical activity experts for the disabled to participate in pro bono work can be analyzed for the revitalization of physical activity for the disabled, the behavior of pro bono participation can also be predicted through the Theory of Planned Behavior.

This study intends to provide basic data for the promotion of physical activity for the disabled by exploring and analyzing factors related to the intention toward participating in pro bono physical activity work for the disabled by pre-service physical activity instructors who are currently majoring in adapted physical activity at universities. Therefore, the purpose of this study is to develop and validate a scale for predicting the behavioral intention of physical activity instructors for people with disabilities to participate in pro bono work based on the Theory of Planned Behavior.

## 2. Materials and Methods

### 2.1. Participants

The selected participants of the study were 322 students who are currently majoring in adapted physical activity at universities in South Korea. The reason why university students were selected as the participants of this study is that they are preliminary physical activity instructors for the disabled. Furthermore, it is necessary to understand the intention of pre-service physical activity instructors for the disabled toward participating in pro bono work because future pro bono plans can be predicted. The demographic characteristics of participants in this study are outlined in Table 1.

### 2.2. Research Instrument

In order to develop a scale to explore and analyzing the intentions of pre-service physical activity instructors regarding pro bono participation, a survey which targeted university students from the department of adapted physical activity was conducted regarding the intention of participating in pro bono work for the promotion of physical activity for the disabled. A questionnaire was prepared based on Ajzen’s “Constructing a theory of planned behavior questionnaire” [28]. In particular, in order to analyze the intention toward participating in pro bono work as accurately as possible, the question content and method were established based on the Theory of Planned Behavior and previous studies, and the final question was validated through consultation with experts. Based on the variables of the pre-service physical activity instructor’s participation intention with respect to pro bono work derived through these questionnaires and expert opinions, a measure of the pre-service physical activity instructor’s pro bono participation intention for promoting physical activity for the disabled was developed.

The Theory of Planned Behavior used in the questionnaire to develop the measure of intention for pre-service physical activity instructors to participate in pro bono work is a total of four variables: intention, behavioral belief, normative belief, and control belief. Among the variables of the questionnaire, the intention was a measure of “intention toward participating in Pro Bono”, and the behavioral belief was a measure of “the degree of belief that certain outcomes may occur due to behavior in Pro Bono participation”. The normative belief was a measure of “perceived groups and people with respect to Pro Bono behavior”, and the control belief was a measure of “helping and hindering the practice of Pro Bono behaviors” [27,28]. In addition, this questionnaire used a 5-point Likert scale (e.g., 1 = strongly agree, 2 = agree, 3 = neither disagree or agree, 4 = disagree, and 5 = strongly disagree).

### 2.3. Research Process

In order to develop a questionnaire on the intention of pre-service physical activity instructors toward participating in pro bono work for the promotion of sports for the disabled, analysis of the literature and prior research were first investigated. Based on the collected data, modifications were made to fit Ajzen’s sample questionnaire [28], and the pilot questionnaire was completed after review by 5 experts who study in the field of adapted physical activity and are faculty members at universities in South Korea. The completed questionnaire was distributed to 100 pre-service physical activity instructors, the collected data were used to evaluate the validity and reliability of the questionnaire, and the results were derived. Another expert meeting was held to discuss the derived results, and the questionnaire for this study was distributed to the study participants based on the completed questionnaire. This process was verified through the institutional review board (IRB) approval of Korea Nazarene University, and the questionnaire scale was developed for this study through the questionnaire validity and reliability test process based on the collected data.

### 2.4. Data Analysis

In this study, the exploratory factor analysis (EFA) was performed using SPSS 21.0 to confirm the validity of the measurement tool. The maximum likelihood method was used for factor extraction, and square rotation was used for factor rotation. To verify the relationship between latent and observed variables, confirmatory factor analysis (CFA) was performed using AMOS 21.0, and the significance level was set to α = 0.05. The maximum likelihood method was used for the estimation method, and x2, Q, the root mean square residual (RMR), Tucker–Lewis index (TLI), comparative fit index (CFI), and normed fit index (NFI) were used for the selection of goodness-of-fit. In addition, conceptual reliability and mean variance extraction were analyzed in the process of confirming factor analysis to verify the reliability of internal consistency. Cronbach’s alpha was used for the internal reliability analysis, and the significance level was set as α = 0.05. Cronbach’s alpha required 0.8 or higher for reliability.

## 3. Results

### 3.1. The Results of the EFA of the Pilot Questionnaire

The data of the pilot questionnaire for developing and validating a scale for predicting the pro bono participation intention of physical activity instructors for people with disabilities were randomly collected from 100 pre-service physical activity instructors in South Korea. The researchers conducted a series of EFA using SPSS 21.0 to identify the structure of the pilot questionnaire. The variables, measure of sampling adequacy, correlation matrix, and adequacy of factor analysis were calculated by the Kaiser-Meyer-Olkin (KMO) and Bartlett’s Test of Sphericity [29]. In the EFA of the pilot questionnaire, the KMO for the measure of sampling adequacy was 0.935. A KMO value higher than 0.6 is considered good [30]. Bartlett’s Test of Sphericity was significant (Bartlett’s χ^2^ = 6955.134, df = 325, *p* < 0.001). In addition, the 68.625% of total variance was totally explained by 4 factors. The communality of retained questions ranged from 0.234 to 0.856 and the loadings of retained question ranged from −0.497 to 0.890. The variance of each factor was calculated as intention (16.534%), behavioral belief (18.636%), normative belief (18.667%), and control belief (14.788%). As a result, the questions of the communality or loading of less than 0.40 (e.g., Q1, Q25) and high multicollinearity were eliminated to improve the structure of the questionnaire. Therefore, the variables, measure of sampling adequacy, correlation matrix, and adequacy of factor analysis were appropriate to determine the purpose of this study. However, the final EFA, CFA, and internal reliability were conducted to enhance the factor structure (e.g., factor loading, internal consistency of factors) of the questionnaire. The results of the EFA of the pilot questionnaire are in Table 2.

### 3.2. The Results of the EFA of the Final Questionnaire

The data in the final questionnaire were collected from 322 pre-service physical activity instructors in South Korea to develop and validate a scale for predicting the pro bono participation intention of physical activity instructors for people with disabilities. The final EFA was conducted to determine the structure of the final questionnaire. In the EFA of the final questionnaire, the KMO for the measure of sampling adequacy was 0.928. Bartlett’s Test of Sphericity was significant (Bartlett’s χ^2^ = 5494.070, df = 153, *p* < 0.001). Further, the 80.312% of total variance was totally explained by 4 factors. The communality of retained questions ranged from 0.612 to 0.880 and the loadings of retained question ranged from 0.539 to 0.887 in all factors. The variance of each factor was calculated as intention (18.115%), behavioral belief (24.358%), normative belief (23.271%), and control belief (14.568%). Cronbach’s alpha coefficients for intention, behavioral belief, normative belief, and control belief were 0.940, 0.948, 0.938, and 0.856. Therefore, the variables, measure of sampling adequacy, correlation matrix, and adequacy of factor analysis were improved and appropriate to identify the purpose of this study. The results of the EFA of the final questionnaire are in Table 3.

### 3.3. The Results of the CFA

A series of CFA were conducted to identify a relationship between the theory of planned behavior and variables in this study to support the structure of the questionnaire based on the relevant theory. In addition, the CFA was used to identify the adequacy of the model fit to the data. In the first CFA, the indexes of RMR, TLI, CFI, and NFI were calculated as RMR (0.051), TLI (0.912), CFI (0.923), and NFI (0.897). According to the researchers (30–32), the value of 0.05 or lower for RMR and the value of 0.09 or higher for TLI, CFI, and NFI were stated as a good model fit. However, the indexes of RMR and NFI did not meet this standard. As a result, six questions (Q2, Q7, Q18, Q20, Q21, Q26) were eliminated due to the low values of these questions in the squared multiple correlation (SMC), high multicollinearity, and modification index. In the final CFA, the indexes of RMR, TLI, CFI, and NFI were improved as follows: RMR (0.038), TLI (0.929), CFI (0.940), and NFI (0.919). The results of model fit of the series of CFA are in Table 4. The diagram of results of the measurement model is in Figure 1. Furthermore, all questions for each factor were measured to be significant (e.g., *p* < 0.001) based on the results of CFA. The results of CFA are in Table 5. Therefore, these index values can be predictable for the adequacy of validity of scale in the questionnaire and model fit.

### 3.4. The Questions for Each Factor in the Final Questionnaire

The final questionnaire was developed and validated after the series of EFA and CFA to predict the behavioral intention of physical activity instructors for people with disabilities to participate in pro bono work based on the Theory of Planned Behavior. The questions for each factor in the final questionnaire are in Table 6.

## 4. Discussion

The purpose of this study was to develop and validate the scale for predicting the pro bono participation intention of pre-service physical activity instructors for persons with disabilities based on the Theory of Planned Behavior. Results were obtained relating to questionnaire scale development and validation for pre-service physical activity instructors for persons with disabilities participating in pro bono work. First, the questionnaire about “Intention Toward Participating in Pro Bono of Pre-service Physical Activity Instructors for the Activation of Physical Activity for the Disabled” was validated based on the Theory of Planned Behavior. In addition, there were four results based on the questionnaire; first, most participants in this study agreed to participate in pro bono work for the physical activity of people with disabilities, and the negative questions about Pro Bono participation have been eliminated due to disagreement of participants. This may be because the all participants in this study are majoring in an area closely related to physical activity for the disabled. Second, the teaching experience about physical activity for the disabled and the knowledge about physical activity for the disabled were required to participate in pro bono work for the physical activity of people with disabilities. Third, some people (e.g., parents of people with disabilities, people with disabilities, family members, friends, students in my department) may believe that pre-service physical activity instructors for the disabled should participate in pro bono work for the physical activity of people with disabilities. Fourth, some elements (e.g., the state of mind of physical activity instructors for people with disabilities, the ability to create an IEP, the ability to do physical activity) can be encouraged to control pro bono participation. The discussion based on the results is as follows:

First, it suggested that this questionnaire based on the Theory of Planned Behavior was appropriate to identify pre-service physical activity instructors’ intention toward participating in pro bono work for people with disabilities. This questionnaire was categorized by Ajzen’s “Constructing a theory of planned behavior questionnaire” [28,29,30,31,32]. Ajzen’s guideline made it appropriate and valid to create the questionnaire. Furthermore, many studies related to physical activity instructors for persons with disabilities based on the Theory of Planned Behavior were conducted for development and validity of a scale measuring their behavioral intention. For instance, Jeong and Block [33] investigated general physical education teachers’ intentions to teach students with disabilities. Lee, Yun, So [25], Rizzo, So, and Tripp [34], and Tripp and Rizzo [35] determined physical education teachers’ intentions to teach people with disabilities. These researchers reported that the questionnaire used in their studies had an applicable and valid scale to determine the purpose of their studies. Eventually, it was possible to confirm their valid and reliable results, even though the purpose of the study was different, as stated by the previous studies related to this study.

Second, in regard to behavioral belief, teaching experience and knowledge about people with disabilities can help instructors participate in pro bono work for the physical activity of people with disabilities. It can be correlated to the results of a study [35] which found that physical education teachers who had teaching experience with students with disabilities had higher intentions to teach disabled students than those with no teaching experience in an inclusive physical education environment. In particular, many researchers reported that experiences with students with disabilities are significant for successful physical activity [36,37]. In addition, one study stated that pre-service physical education teachers with more education (e.g., undergraduate courses about children with disability) and information about students with disabilities had a positive attitude toward teaching students with disabilities [36]. Hence, it is deduced that experience with and knowledge of teaching the disabled are important factors in behavioral belief toward participating in pro bono work regarding physical activity of the disabled. Furthermore, it is determined that a study on whether these factors lead to actual behavior is needed.

Third, parents of people with disabilities, people with disabilities, family members, friends, and students in my department were the people who had a lot of influence on my participation in pro bono work for the physical activity of people with disabilities. The participants of this study were all university students studying as pre-service physical activity instructors. University students show high interest in human rights and social issues such as topics around persons with disabilities. [38]. In particular, one study [39] reported that many university students argue that equal opportunities should be provided to persons with disabilities, and this argument is evidence that students are sensitive to social pressures (i.e., demand of parents of people with disabilities or people with disabilities). In addition, some researchers [40] reported that parents and friends of university students play the most important roles in determining the behavior of university students who are in early adulthood, as they understand and help university students. Thus, these factors (i.e., parents of people with disabilities, people with disabilities, family members, friends, students in my department) must have had a lot of influence on the behavior of the pre-service physical activity instructors to participate in pro bono work, and it is necessary to find out what kind of influence they had.

Fourth, the state of mind of physical activity instructors for people with disabilities, the ability to create the IEP, and the ability to do physical activity can help to control the pro bono participation of physical activity instructors for people with disabilities. In the Theory of Planned Behavior, there is a correlation between each factor, and control belief and intention also have a correlation [26,27,28]. A willingness to do something can be interpreted as a person’s state of mind and determination. Consequently, the control belief may be different depending on the person’s mindset. The IEP is an educational program that comprises all areas of education, including educational goals, teaching, and evaluation for the education of students with disabilities based on the abilities of individual students with disabilities [41,42]. Furthermore, physical activity instructors can teach well if they know how to perform the physical activity they need to teach. These abilities (e.g., the ability to create the IEP and to perform physical activity) can increase the self-efficacy of instructors, and self-efficacy can improve self-confidence and control their behavior [43]. Then, it is necessary to find ways to improve these elements (e.g., the state of mind of physical activity instructors for people with disabilities, the ability to create an IEP, and the ability to do physical activity) to control successful behavior.

## 5. Research Limitations

There are some limitations. First, this study was conducted only by university students majoring in adapted physical activity, so it may be difficult to apply it in the actual field. Thus, future research should be conducted for physical activity instructors in the actual field. Second, this study was conducted only with participants in South Korea, so there may be cultural differences. Hence, it is necessary to expand the criteria for participants in future studies. Third, this study was conducted targeting pre-service physical activity instructors. It was not possible to know the actual participation in pro bono work. Consequently, in future studies, it is necessary to check whether the behavior was actually performed through a longitudinal study.

## 6. Conclusions

This study developed and validated a scale for predicting the pro bono participation intention of physical activity instructors for people with disabilities based on the Theory of Planned Behavior. The conclusions are as follows: first, the questionnaire developed in this study was validated based on the Theory of Planned Behavior. Second, the behavioral belief of participants was influenced by their teaching experience in physical activity for the disabled and knowledge about physical activity for the disabled. Third, the normative belief was influenced by the parents of people with disabilities, people with disabilities, family members, friends, and students in my department. Fourth, the control belief was influenced by the state of mind of physical activity instructors for people with disabilities, the ability to create an IEP, and the ability to do physical activity.

## Figures and Tables

**Figure 1 healthcare-10-02094-f001:**
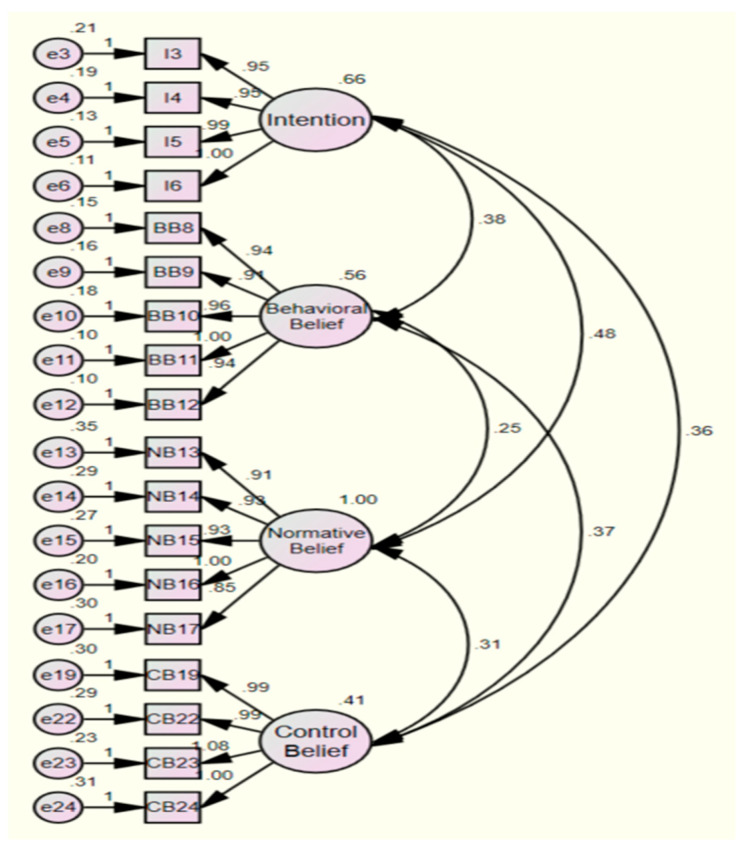
The diagram of results of the measurement model.

**Table 1 healthcare-10-02094-t001:** Demographic characteristics of participants.

Classification	Content	N	Percentage (%)
Gender	Male	209	64.9
	Female	113	35.1
Grade	Freshman	34	10.6
	Sophomore	107	33.2
	Junior	115	35.7
	Senior	66	20.5

**Table 2 healthcare-10-02094-t002:** The results of the EFA of the pilot questionnaire.

No	Factors				Communality
	Intention	Behavioral Belief	Normative Belief	Control Belief	
Q1	0.199	0.034	0.890	0.131	0.324
Q2	0.103	0.079	0.873	0.176	0.790
Q3	0.180	0.069	0.853	0.163	0.789
Q4	0.148	0.118	0.831	0.167	0.809
Q5	0.208	0.171	0.807	0.236	0.856
Q6	0.219	0.263	0.638	0.302	0.839
Q7	0.269	0.849	0.127	0.194	0.505
Q8	0.252	0.829	0.128	0.234	0.820
Q9	0.166	0.811	0.099	0.330	0.806
Q10	0.256	0.799	0.076	0.331	0.790
Q11	0.312	0.775	0.157	0.258	0.849
Q12	0.286	0.617	0.146	0.143	0.824
Q13	0.805	0.287	0.296	0.191	0.755
Q14	0.803	0.283	0.147	0.203	0.812
Q15	0.775	0.121	0.373	0.232	0.792
Q16	0.770	0.273	0.308	0.159	0.852
Q17	0.761	0.283	0.360	0.221	0.781
Q18	−0.497	−0.271	0.057	−0.018	0.617
Q19	0.149	0.129	0.161	0.710	0.613
Q20	0.234	0.270	0.191	0.696	0.433
Q21	0.266	0.326	0.145	0.660	0.571
Q22	0.278	0.416	0.115	0.641	0.634
Q23	−0.041	0.057	0.234	0.602	0.676
Q24	0.178	0.133	0.208	0.582	0.650
Q25	0.342	0.400	0.216	0.537	0.234
Q26	−0.013	0.270	0.042	0.398	0.423
Total	4.299	4.845	4.853	3.845	
%Variance	16.534	18.636	18.667	14.788	
%Cumulative	16.534	35.170	53.837	68.625	

**Table 3 healthcare-10-02094-t003:** The results of the EFA of the final questionnaire.

No	Factors				Communality
	Intention	Behavioral Belief	Normative Belief	Control Belief	
Q3	0.791	0.314	0.259	0.158	0.818
Q4	0.805	0.160	0.332	0.235	0.841
Q5	0.823	0.314	0.241	0.211	0.880
Q6	0.789	0.307	0.309	0.241	0.871
Q8	0.209	0.815	0.073	0.332	0.825
Q9	0.122	0.825	0.108	0.323	0.812
Q10	0.291	0.791	0.137	0.262	0.800
Q11	0.249	0.878	0.110	0.175	0.877
Q12	0.226	0.869	0.124	0.198	0.862
Q13	0.158	0.136	0.856	0.123	0.793
Q14	0.142	0.089	0.887	0.130	0.832
Q15	0.216	0.082	0.851	0.138	0.798
Q16	0.246	0.054	0.884	0.100	0.856
Q17	0.261	0.174	0.785	0.202	0.757
Q19	0.290	0.423	0.238	0.539	0.612
Q22	0.202	0.318	0.163	0.735	0.710
Q23	0.225	0.393	0.122	0.732	0.758
Q24	0.167	0.252	0.222	0.782	0.754
Total	3.261	4.384	4.189	2.622	
%Variance	18.115	24.358	23.271	14.568	
%Cumulative	18.115	42.473	65.744	80.312	
Cronbach α	0.940	0.948	0.938	0.856	

**Table 4 healthcare-10-02094-t004:** The results of model fit index of a series of CFA.

Classification	Goodness of Fit		
	Results of the First CFA		Results of the Final CFA
RMR	0.051	→	0.038
TLI	0.912	→	0.929
CFI	0.923	→	0.940
NFI	0.897	→	0.919

**Table 5 healthcare-10-02094-t005:** The results of CFA.

Classification	Estimate	S.E.	C.R.
Intention → Q3	0.951	0.040	23.714 ***
Intention → Q4	0.952	0.039	24.484 ***
Intention → Q5	0.994	0.035	28.034 ***
Intention → Q6	1.000		
Behavioral Belief → Q8	0.939	0.038	24.796 ***
Behavioral Belief → Q9	0.908	0.038	23.654 ***
Behavioral Belief → Q10	0.963	0.040	23.921 ***
Behavioral Belief → Q11	1.000		
Behavioral Belief → Q12	0.942	0.034	27.702 ***
Normative Belief → Q13	0.914	0.042	21.823 ***
Normative Belief → Q14	0.928	0.040	23.348 ***
Normative Belief → Q15	0.933	0.039	23.829 ***
Normative Belief → Q16	1.000		
Normative Belief → Q17	0.847	0.039	21.782 ***
Control Belief → Q19	0.989	0.074	13.410 ***
Control Belief → Q22	0.993	0.073	13.511 ***
Control Belief → Q23	1.083	0.074	14.624 ***
Control Belief → Q24	1.000		

Note. S.E. = standard error, C.R. = construct reliability, *** *p* < 0.001.

**Table 6 healthcare-10-02094-t006:** The questions for each factor in the final questionnaire.

Factor	No	Question
Intention	Q3	I strongly agree to participate in Pro Bono for the physical activity of people with disabilities.
	Q4	I will definitely participate in Pro Bono for the physical activity of people with disabilities.
	Q5	I will gladly participate in Pro Bono for the physical activity of people with disabilities.
	Q6	I will actively participate in Pro Bono for the physical activity of people with disabilities.
Behavioral Belief	Q8	Participating in Pro Bono for the physical activity of people with disabilities will require the teaching experience about physical activity for the disabled.
	Q9	Participating in Pro Bono for the physical activity of people with disabilities will require the theoretical learning about physical activity for the disabled that I have learned so far.
	Q10	My time effort to participate in Pro Bono for the physical activity of people with disabilities will be worthwhile.
	Q11	Physical activity coaching experience for people with disabilities will be valuable to participate in Pro Bono for physical activity.
	Q12	The knowledge of physical activity for people with disabilities will be valuable to participate in Pro Bono for physical activity.
Normative Belief	Q13	Parents of people with disabilities will think that I should participate in Pro Bono for the physical activity of people with disabilities.
	Q14	People with disabilities will think that I should participate in Pro Bono for the physical activity of people with disabilities.
	Q15	My family members will think that I should participate in Pro Bono for the physical activity of people with disabilities.
	Q16	My friends would think that I should participate in Pro Bono for the physical activity of people with disabilities.
	Q17	Students in my department will think that I should participate in Pro Bono for the physical activity of people with disabilities.
Control Belief	Q19	Physical activity instructors for people with disabilities must be able to participate in Pro Bono, which provides physical activity services for people with disabilities.
	Q22	Participation in Pro Bono providing physical activity services for people with disabilities depends on the mind of physical activity instructors for people with disabilities.
	Q23	Physical activity instructors for people with disabilities must be able to create an IEP (Individualized Education Plan) to participate in Pro Bono, which provides physical activity services for people with disabilities.
	Q24	Physical activity instructors for people with disabilities must have the ability to do physical activity to participate in Pro Bono, which provides physical activity services for people with disabilities.

## Data Availability

The original contributions presented in the study are included in the article. Further inquiries can be directed to the corresponding author.

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
