# Peer review of "Development and Validation of a Scale Measuring Intention toward Participating in Pro Bono of Pre-Service Physical Activity Instructors for the Activation of Physical Activity for the Disabled: Based on the Theory of Planned Behavior"

_healthcare, 2022, doi:10.3390/healthcare10102094_

Round 1

Reviewer 1 Report

El objeto de estudio de este manuscrito no es uno de los temas mas extenso en la bilbiografia científica sobre actividad física. Aunque se ha descrito mucho sobre actividad física para personas con discapacidad, apenas hay informción sobre los procesos que hacen que los técncios de actividad física buscan desarrollar su actividad en este ámbito y mas si es en una situación pro Bono.

Es evidente que el voluntariado en este tipo de servicios está muy extendido, especialmente en Europa.

Recomiendo la publicación de este manuscrito. El metodo de validación de las herramientas es el correcto, asi como la metodología de investigación realizada.

Author Response

We would like to thank Academic reviewer for a thorough review of our manuscript. 

Sincerely,

Kyungjin Kim.

Reviewer 2 Report

In the introduction and in the discussion of results, keep in mind that there is research on other geographical areas that may be interesting.

These deal with the perception of higher education students towards disability or personal skills for working with people with disabilities.

Author Response

(The authors gave the same response as above.)

Reviewer 3 Report

Dear authors,

First of all, I would like to congratulate you on the contribution of this work to the field of study. Next, I would like to make a number of suggestions to improve the work:

- In Table 1 there is an error in the percentage of women. The number is incorrect.
- When talking about the group of experts, it is necessary to be more precise. How many members make up the group, what were the criteria for considering them experts and what are their areas of expertise.
- The level of internal consistency of the questionnaire provided by Cronbach's alpha is poor. I suggest eliminating the questions that lower the level of consistency and recalculate.
- It is not clear to me how the variables referring to intention or behaviour are measured. I think it would be necessary to go more deeply into these aspects, explaining clearly how the data have been obtained and how they have been analysed to construct the answers of the questionnaire.
- Figure 1 has very little resolution. I suggest improving its resolution to make it easier to read.

I hope you find these suggestions useful and that you understand that they are made with the sole intention of contributing to improve your work.

Best regards,

Author Response

We would like to thank Academic reviewer for a thorough review of our manuscript. We believe we have addressed all comments and edits. In our manuscript, all comments and edits are in red words.

  • In Table 1 there is an error in the percentage of women. The number is incorrect. (Thank you for your review. We have modified the percentage of women in Table 1.)
  •  When talking about the group of experts, it is necessary to be more precise. How many members make up the group, what were the criteria for considering them experts and what are their areas of expertise. (Thank you for your review. We have added explanation of experts.)
  • The level of internal consistency of the questionnaire provided by Cronbach's alpha is poor. I suggest eliminating the questions that lower the level of consistency and recalculate. (Thank you for your review. If the level of Crobach's alpha is 0.8 or more, it is significant. In this article, there were the level of Crobach's alpha as 0.8 or more)
  • It is not clear to me how the variables referring to intention or behaviour are measured. I think it would be necessary to go more deeply into these aspects, explaining clearly how the data have been obtained and how they have been analysed to construct the answers of the questionnaire. (Thank you for your review. We have added the variable's explanation of intention toward participating in Pro Bono.)
  • Figure 1 has very little resolution. I suggest improving its resolution to make it easier to read. (Thank you for your review. We have modified Figure 1.)

Reviewer 4 Report

Review Development and Validation of a Scale Measuring In-tention of Participating in Pro Bono of Pre-service Adapted Physical Ac-tivity Instructors for the Activation of Physical Education for Individu-als with Disabilities: Based on the Theory of Planned Behavior. 

 Thank you for the opportunity to review the manuscript entitled Development and Validation of a Scale Measuring Intention Toward Participating in Pro Bono of Pre-service Physical Activ-ity Instructors for the Activation of Physical Activity for the Disabled: Based on the Theory of Planned Behavior  I have read the content of the manuscript with due attention, and although I appreciate both the idea of the research, as well as its implementation and preparation of the results, I regret to say that it is not suitable in its current form for publication. A far-reaching revision is necessary, but towards rewriting the article rather than improving the current text. First of all, the article is awkwardly worded, the order, syntax, logical structure of many (most) sentences make reading difficult to read eg. “In order to develop a scale by exploring and analyzing the intentions of pre-service physical activity instructors regarding Pro Bono participation, a survey was conducted on the intention of participating in Pro Bono for the activation of physical activity for the disabled targeting university students of department of adapted physical activity”. It is absolutely necessary that the text undergoes a thorough proofreading by a native speaker or at least an English teacher.

The Introduction is too long and full of inaccuracies - some of which are probably a derivative of the aforementioned little communicative language which the work was written. As an example of inaccuracies: “… In the case of chronic diseases, the prevalence of hypertension in the non-disabled age group was 33.5%” – what age group?

Exercise maintains and improves physical strength and body functions through physical training, so it can be effective for health management as well as disease prevention - I don't understand how exercise is supposed to affect the body through physical training? I have the impression that the authors do not consistently use the terms exercise and training. Staying with this type of statement, it is impossible to accept the reductionist approach presented by the authors - exercise and physical strength; and what about other forms of physical activity that are not de facto exercises, and what about many other effects of this activity, e.g. in the cognitive or emotional sphere. It would be more logical to list the best-documented health effects of physical activity, including those conditions that it can prevent and / or support treatment, than to limit ourselves to a casual and not entirely precise statement. in the next sentence, the authors refer to physical education, which is an incomprehensible contextual shift.

Looking at the Introduction as a whole, it is too long - some sentences and entire paragraphs are unnecessary and the theoretical justification of the research is not convincing. The latter simply boils down to a short description of the CBT assumptions - in a way that does not fully reflect the essence of this theory - but why it should constitute a good theoretical basis for understanding the subject of the authors' research, we do not find out.

There is some error in table 1 - what the value 4135.1% means?

The description of the questionnaire lacks information about the scale in which the items were assessed - was it a semantic difference (nota bene recommended by Ajzen), Likert's scale or maybe another one? An equally important lack is insufficient information on how the items were assessed by competent judges (what evaluation method was used?) And their competences in the field of TPB (it is only mentioned that they were “adapted physical activity experts”).

In the sentence regarding the Cronbach's alpha coefficient, there is no information about the prag which was taken as the limit of reliability (although this was rather high, as can be judged from Table 3).

I consider it unnecessary accuracy to enter values up to 3 decimal places (eg. 18.636%) - except for the p value it is absolutely redundant.

My serious doubt is the compliance of the content of the items with the substantive content of the constructs that are to be measured by them. this is especially true of the Control belief subscale. Reading the content of the items, we see postulative statements reflecting the beliefs of the respondent about a certain characteristic of people - activity instructors. Meanwhile, the construct of perceived behavioral control is, according to the author of the theory, to reflect an individual's subjective judgments regarding her or his capacity and ability to engage in specific behavior. Hence, such ways of formulating items as “how much control do you have over …” or “I believe I have the ability to …”

Author’s wrote that “it suggested that this questionnaire based on the Theory of Planned Behavior was appropriate to identify pre-service physical activity instructor’s intention toward participating in Pro Bono for people with disabilities”.

I do not fully understand the real intention of the authors, especially when confronted with the essence of the theory. If the goal is to get to know the instructors' intentions, then be expeditious just ... ask about those intentions. Rather, TPB was created as an attempt to understand the factors underlying certain behaviors - which direct predictor is behavioral intention.

I do not fully understand the real intention of the authors, especially when confronted with the essence of the theory. If the goal is to get to know the instructors' intentions, then be expeditious just ... ask about those intentions. Rather, CBT was created as an attempt to understand the factors underlying certain behaviors - whose direct predictor is behavioral intention. I would see the sense of developing a questionnaire such as the one described in this article in identifying the significance of factors that may predict pre-service physical activity instructor’s intentions, and not in diagnosing the intentions themselves - for the authors to think about.

Taking into account the reservations presented, they do not see the possibility of recommending the article for publication in its present form.

Author Response

We would like to thank Academic reviewer for a thorough review of our manuscript. We believe we have addressed all comments and edits. In our manuscript, all comments and edits are in red words.

  • The Introduction is too long and full of inaccuracies - some of which are probably a derivative of the aforementioned little communicative language which the work was written. As an example of inaccuracies: “… In the case of chronic diseases, the prevalence of hypertension in the non-disabled age group was 33.5%” – what age group? (Thank you for your review. We have modified it in the article.)

  • Exercise maintains and improves physical strength and body functions through physical training, so it can be effective for health management as well as disease prevention - I don't understand how exercise is supposed to affect the body through physical training? I have the impression that the authors do not consistently use the terms exercise and training. Staying with this type of statement, it is impossible to accept the reductionist approach presented by the authors - exercise and physical strength; and what about other forms of physical activity that are not de facto exercises, and what about many other effects of this activity, e.g. in the cognitive or emotional sphere. It would be more logical to list the best-documented health effects of physical activity, including those conditions that it can prevent and / or support treatment, than to limit ourselves to a casual and not entirely precise statement. in the next sentence, the authors refer to physical education, which is an incomprehensible contextual shift. (Thank you for your review. We have modified it from physical education to physical activity.)

  • Looking at the Introduction as a whole, it is too long - some sentences and entire paragraphs are unnecessary and the theoretical justification of the research is not convincing. The latter simply boils down to a short description of the CBT assumptions - in a way that does not fully reflect the essence of this theory - but why it should constitute a good theoretical basis for understanding the subject of the authors' research, we do not find out. (Thank you for your review. We have modified it in the article.)

  • There is some error in table 1 - what the value 4135.1% means? (Thank you for your review. We have modified it.)

  • The description of the questionnaire lacks information about the scale in which the items were assessed - was it a semantic difference (nota bene recommended by Ajzen), Likert's scale or maybe another one? An equally important lack is insufficient information on how the items were assessed by competent judges (what evaluation method was used?) And their competences in the field of TPB. (Thank you for your review. We have modified it in the article.)

  • In the sentence regarding the Cronbach's alpha coefficient, there is no information about the prag which was taken as the limit of reliability. (Thank you for your review. We have added it in the article.)

  • I consider it unnecessary accuracy to enter values up to 3 decimal places (eg. 18.636%) - except for the p value it is absolutely redundant. (Thank you for your review.)

  • My serious doubt is the compliance of the content of the items with the substantive content of the constructs that are to be measured by them. this is especially true of the Control belief subscale. Reading the content of the items, we see postulative statements reflecting the beliefs of the respondent about a certain characteristic of people - activity instructors. Meanwhile, the construct of perceived behavioral control is, according to the author of the theory, to reflect an individual's subjective judgments regarding her or his capacity and ability to engage in specific behavior. Hence, such ways of formulating items as “how much control do you have over …” or “I believe I have the ability to …”

    Author’s wrote that “it suggested that this questionnaire based on the Theory of Planned Behavior was appropriate to identify pre-service physical activity instructor’s intention toward participating in Pro Bono for people with disabilities”.

    I do not fully understand the real intention of the authors, especially when confronted with the essence of the theory. If the goal is to get to know the instructors' intentions, then be expeditious just ... ask about those intentions. Rather, TPB was created as an attempt to understand the factors underlying certain behaviors - which direct predictor is behavioral intention. (Thank you for your review. We have added it in the article.)

    I do not fully understand the real intention of the authors, especially when confronted with the essence of the theory. If the goal is to get to know the instructors' intentions, then be expeditious just ... ask about those intentions. Rather, CBT was created as an attempt to understand the factors underlying certain behaviors - whose direct predictor is behavioral intention. I would see the sense of developing a questionnaire such as the one described in this article in identifying the significance of factors that may predict pre-service physical activity instructor’s intentions, and not in diagnosing the intentions themselves - for the authors to think about.

Finally, our article is under review in English once more. We will review it again before submitting the final article.

Sincerely,

Kyungjin Kim.

Round 2

Reviewer 4 Report

I still have doubts about the compliance of the research formula with the assumptions of TPB, but I am willing to leave the evaluation of the authors' work to the readers. Undoubtedly, the authors made a number of corrections that corrected many of the shortcomings noted earlier.  Therefore, I am inclined to accept the current version of the article